# Benchmarking Stochastic Approximation Algorithms for Fairness-Constrained Training of Deep Neural Networks

## Abstract

The ability to train Deep Neural Networks (DNNs) with constraints is instrumental in improving the fairness of modern machine-learning models. Many algorithms have been analysed in recent years, and yet there is no standard, widely accepted method for the constrained training of DNNs. In this paper, we provide a challenging benchmark of real-world large-scale fairness-constrained learning tasks, built on top of the US Census (Folktables, [22]). We point out the theoretical challenges of such tasks and review the main approaches in stochastic approximation algorithms. Finally, we demonstrate the use of the benchmark by implementing and comparing three recently proposed, but as-of-yet unimplemented, algorithms both in terms of optimization performance, and fairness improvement. We release the code of the benchmark as a Python package at https://github.com/humancompatible/train.

## 1 Introduction

There has been a considerable interest in detecting and mitigating bias in artificial intelligence (AI) systems, recently. Multiple legislative frameworks, including the AI Act in the European Union, require the bias to be removed, but there is no agreement on what the correct definition of bias is or how to remove it. A natural translation of the requirement of removing bias into the formulation of training of deep neural network (DNN) utilizes constraints bounding the difference in empirical risk across multiple subgroups [13, 42, 48]. Over the past five years, there have been numerous algorithms ([29, 5, 17, 43, 6, 40, 41, 10, 18, 49, 27, 32, 33]) proposed to solve convex and non-convex empirical-risk minimization (ERM) problems subject to constraints bounding the absolute value of empirical risk. Numerous other algorithms of this kind could be construed, based on a number of design choices, including:

- sampling techniques for the ERM objective and the constraints, either the same or different;
- use of first-order or higher-order derivatives, possibly in quasi-Newton methods;
- use of globalization strategies such as filters or line search;
- use of "true" globalization strategies including random initial points and random restarts in order to reach global minimizers.

Nevertheless, there is no single toolkit implementing the algorithms, which would allow for their easy comparison, and there is no benchmark to test the combinations of design choices on.

In this paper, we consider the constrained ERM problem:

$$\min_{x \in \mathbb{R}^n} \mathbb{E}[f(x, \xi)] \quad \text{s.t.} \quad \mathbb{E}[c(x, \zeta)] \leq 0, \tag{1}$$

Table 1: Particular formulations of the constraint function $c$ to enforce fairness.

| Model | Our formulation |
|---|---|
| Demographic Parity [24] | $\|\mathbb{E}_{\mathcal{D}[\text{group } A]}[\ell(f_\theta(X), Y)] - \mathbb{E}_{\mathcal{D}[\text{group } B]}[\ell(f_\theta(X), Y)]\| \le \delta$ |
| Equal opportunity [31] | $\|\mathbb{E}_{\mathcal{D}[\text{group } A, Y=+]}[\ell(f_\theta(X), Y)] - \mathbb{E}_{\mathcal{D}[\text{group } B, Y=+]}[\ell(f_\theta(X), Y)]\| \le \delta$ |
| Equalized odds [31] | $\sum_{v \in \{+,-\}} \|\mathbb{E}_{\mathcal{D}[\text{group } A, Y=v]}[\ell(f_\theta(X), Y)] - \mathbb{E}_{\mathcal{D}[\text{group } B, Y=v]}[\ell(f_\theta(X), Y)]\| \le \delta$ |

where $\xi$ and $\zeta$ are random variables. Further, we provide an automated way of constructing the ERM formulations out of a computation graph of a neural network defined by PyTorch or TensorFlow, the choice of the constraints (see Table 1), and a definition of the protected subgroups to apply the constraints to. Specifically, we provide means of utilizing the US Census data via the Python package Folktables, together with definitions of up to 5.7 billion protected subgroups. This presents a challenging benchmark in stochastic approximation for the constrained training of deep neural networks.

**Our contributions.** The contributions of this paper are:

- a literature review of algorithms subject to handling (1);

- a toolbox that *(i)* implements four algorithms applicable in real-world situations, and *(ii)* provides an easy-to-use benchmark on real-world fairness problems;

- numerical experiments that compare these algorithms on a real-world dataset, and a comparison with alternative approaches to fairness.

**Paper structure.** The rest of the paper is organized as follows. Section 2 reviews related works and presents the relevant notions of fairness. Section 3 introduces the algorithms. Section 4 reports on our experiments. Section 5 concludes.

## 2 Related work, and background in fairness

In the literature on fairness, one distinguishes among pre-processing, in-processing, and post-processing. Pre-processing methods focus on modifying the training data to mitigate biases [50, 23]. In-processing methods enforce fairness during the training process by modifying the learning algorithm itself [53]. Post-processing methods adjust the model's predictions after training [35]. The constrained ERM approach (1) belongs to the class of in-processing methods.

In-processing methods include several approaches. One trend consists in jointly learning a predictor function and an adversarial agent that aims to reconstitute the subgroups from the predictor [1, 38, 39, 25]. Another approach consists in adding "penalization" terms to the empirical risk term. These additional penalization terms, commonly referred to as regularizers, promote models that are a compromise between fitting the training data, and optimizing a fairness metric. Differentiable regularizers include, among others, HSIC [37], Fairret [11], or Prejudice Remover [34].

Closer to our setting, [16] considers minimizing the empirical risk subject to the so-called rate constraints based on the model's prediction rates on different datasets. These rates, derived from a dataset, give rise to non-convex, non-smooth, and large-scale inequality constraints akin to (1). The authors of [16] argue that hard constraints, although leading to a more difficult optimization problem, offer advantages over using a weighted sum of multiple penalization terms. Indeed, while the choice of weights for the penalization terms may depend on the dataset, specifying one constraint for each goal is easier for practitioners. In addition, a penalization-based model provides a predictor that balances minimizing the data-fit term and penalties in an opaque way, whereas a constraint-based model allows for a clearer understanding of the model design: minimizing the data-fit term subject to "hard" fairness constraints. Rate constraints differ from those in (1) in that they are piecewise-constant, rendering first-order methods unsuitable for solving them.

Major toolboxes for evaluating the fairness of models or for training models with fairness guarantees include AIF360 [4] and FairLearn [8]. Other libraries include [21], which computes the Pareto front of accuracy and fairness metrics for high-capacity models, and [11], which provides differentiable fairness-inducing penalization terms.

A detailed survey of fairness-oriented datasets is provided in [36], and new datasets are derived in [22]. The benchmark [30] provides a review of the existence of biases in prominent datasets, finding that "not all widely used fairness datasets stably exhibit fairness issues", and assesses the performance of a wide range of in-processing fairness methods in addressing biases, focusing on differentiable minimization only. Other benchmarks of fairness methods include [20, 26, 46, 15]. The statistical aspects of the fairness-constrained Empirical Risk Minimization have only been considered recently; see e.g. [12].

The template problem (1) encompasses fairness-enforcing approaches that find applications in high-risk domains, such as credit scoring, hiring processes, medicine and healthcare [14], ranking and recommendation [47], but also in forecasting the observations of linear dynamical systems [55], or in two-sided economic markets [54]. In addition, solving (1) is of interest in other fields, such as compression of neural networks [13], improving statistical performance of neural networks [42, 48], or the training of neural networks with constraints on the Lipschitz bound [45]. We note that the presence of large-scale constraints is a common feature to all the aforementioned methodologies.

**Deep neural networks (DNNs).** Consider a dataset of $N$ observations $\mathcal{D} = \{(X_i, Y_i), i = 1, ..., N\}$. We seek some function $f_\theta$ such that $f_\theta(X_i) \approx Y_i$. A typical formulation of this task is the following regression problem:

$$\min_{\theta \in \mathbb{R}^n} \frac{1}{N} \sum_{i=1}^{N} \ell(f_\theta(X_i), Y_i) + \mathcal{R}(\theta). \tag{2}$$

Here, $\ell : \mathbb{R} \times \mathbb{R} \to \mathbb{R}$ is a loss function, such as the logistic loss $\ell(y; z) = \log(1 + e^{-yz})$, the hinge loss $\ell(y; z) = \max\{0, 1 - yz\}$, the absolute deviation loss $\ell(y; z) = |y - z|$, or the square loss $\ell(y; z) = \frac{1}{2}(y - z)^2$. The term $\mathcal{R}$ is a regularizer, and $f_\theta$ is a deep neural network (DNN) of depth $L$ with parameters $\theta$. The DNN $f_\theta$ is defined recursively, for some input $X$, as

$$a_0 = X, \qquad a_i = \rho_i(V_i(\theta)a_{i-1}), \ \text{ for every } i = 1, \dots, L, \qquad f_\theta(X) = a_L, \tag{3}$$

where $V_i(\cdot)$ are linear maps into the space of matrices, and $\rho_i$ are activation functions applied coordinate-wise, such as ReLU $\max(0, t)$, quadratics $t^2$, hinge losses $\max\{0, t\}$, and SoftPlus $\log(1 + e^t)$. A dataset $\mathcal{D}$ is described by attributes (or features), such as age, income, gender, etc. The attribute which the DNN is trained to predict is called the class attribute. We denote the class attribute by $Y$, whereas the predicted value given by the DNN is denoted by $\hat{Y}$. Both $Y$ and $\hat{Y}$ are binary and take values in $\{+, -\}$.

**Fairness-aware learning applied to DNNs.** The goal of this approach is to reduce discriminatory behavior in the predictions of a DNN across different demographic groups (e.g., male vs. female). The demographic groups are also reffered to as subgroups. The attributes such as race or gender which must be handled cautiously are called protected. We denote by $S$ the protected attribute $S \in \{s, \bar{s}\}$ where $s$ denotes the protected group and $\bar{s}$ denotes the non-protected group. Denote by $\mathcal{D}[s]$ and $\mathcal{D}[\bar{s}]$ the observations in $\mathcal{D}$ such that $S = s$ and $S = \bar{s}$, respectively. A way to impose fairness on the learned predictor is to equip (2) with suitable constraints. Some possible constraint choices are shown in Table 1. Choosing loss difference bound as the constraint and setting $\delta > 0$ yields formulation:

$$\min_{\theta \in \mathbb{R}^n} \quad \frac{1}{N} \sum_{i=1}^{N} \ell(f_\theta(X_i), Y_i) + \mathcal{R}(\theta)$$
$$\text{s.t.} \quad -\delta \leq \frac{1}{|\mathcal{D}[s]|} \sum_{X_i, Y_i \in \mathcal{D}[s]} \ell(f_\theta(X_i), Y_i) - \frac{1}{|\mathcal{D}[\bar{s}]|} \sum_{X_i, Y_i \in \mathcal{D}[\bar{s}]} \ell(f_\theta(X_i), Y_i) \leq \delta. \tag{4}$$

**Fairness metrics.** There exist tens of fairness metrics [51], however, it was pointed out in [3, Ch. 3] that most fairness metrics may be seen as combinations of independence, separation, and sufficiency. These baseline fairness criteria cannot be attained simultaneously. Moreover, there is a trade-off between attaining the baseline fairness metrics and the prediction accuracy, i.e., the probability that the predicted value is equal to the actual value. As a result, we seek an optimal trade-off between attaining the fairness metrics and minimizing the prediction inaccuracy. We follow the definitions in [3] of the baseline fairness metrics applied to a binary classification task.

**Independence (Ind)**   This fairness criterion requires the prediction $\hat{Y}$ to be statistically independent of the protected attribute $S$. Equivalent definitions of independence for a binary classifier $\hat{Y}$ are referred to as statistical parity (SP), demographic parity, and group fairness. Independence is the simplest criterion to work with, both mathematically and algorithmically. In a binary classification task, independence implies the equality of $P(\hat{Y} = + \mid S = s)$ and $P(\hat{Y} = + \mid S = \bar{s})$ and the fairness gap is computed as

$$|P(\hat{Y} = + \mid S = s) - P(\hat{Y} = + \mid S = \bar{s})|.$$

**Separation (Sp)**   Unlike independence, the separation criterion requires the prediction $\hat{Y}$ to be statistically independent of the protected attribute $S$, given the true label $Y$. The separation criterion also appears under the name Equalized odds (EO). In a binary classification task, the separation criterion requires that all groups experience the same true negative rate and the same true positive rate. Formally, we require the equality of $P(\hat{Y} = + \mid S = s, Y = v)$ and $P(\hat{Y} = + \mid S = \bar{s}, Y = v)$, for every $v \in \{+, -\}$. The fairness gap may be computed as

$$\sum_{v \in \{+, -\}} |P(\hat{Y} = + \mid S = s, Y = v) - P(\hat{Y} = + \mid S = \bar{s}, Y = v)|.$$

**Sufficiency (Sf)**   The sufficiency criterion is satisfied if the true label $Y$ is statistically independent of the protected attribute $S$, given the prediction $\hat{Y}$. In a binary classification task, the sufficiency criterion requires a parity of positive and negative predictive values across the groups. Formally, we require the equality of $P(Y = + \mid \hat{Y} = v, S = s)$ and $P(Y = + \mid \hat{Y} = v, S = \bar{s})$, for every $v \in \{+, -\}$, and the fairness gap may be computed as

$$\sum_{v \in \{+, -\}} |P(Y = + \mid S = s, \hat{Y} = v) - P(Y = + \mid S = \bar{s}, \hat{Y} = v)|.$$

# 3   Algorithms

We recall that we consider the optimization problem

$$\min_{x \in \mathbb{R}^n} F(x) \quad \text{s.t.} \quad C(x) \leq 0, \tag{5}$$

where the functions $F : \mathbb{R}^n \to \mathbb{R}$ and $C : \mathbb{R}^n \to \mathbb{R}^m$ are defined as expectations of functions $f$ and $c$, which depend on random variables $\xi$ and $\zeta$, respectively. Solving (5) has the following challenges:

- large-scale objective and constraint functions, which require sampling schemes,
- the necessity of incorporating inequality constraints, not merely equality constraints (see fairness formulations in Table 1),
- the necessity to cope with the nonconvexity and nonsmoothness of $F$ and $C$, due to the presence of neural networks.

In this section, we identify the algorithms that address these challenges most precisely. However, we note that there exists currently no algorithm with guarantees for such a general setting.

**Recalls and notation.**   We denote the projection of a point $x$ onto a set $\mathcal{X}$ by $\text{proj}_{\mathcal{X}}(x) = \arg\min_{v \in \mathcal{X}} \|x - v\|^2$. We denote by $N \sim \mathcal{G}(p_0)$ sampling a random variable from the geometric distribution with a parameter $p_0$, i.e., the probability that $N = n$ equals $(1 - p_0)^n p_0$ for $n \geq 0$. We distinguish between the random variable $\xi$ associated with the objective function and the random variable $\zeta$ associated with the constraint function. Their probability distributions are denoted by $\mathcal{P}_\xi$ and $\mathcal{P}_\zeta$. For an integer $J \in \mathbb{N}$, a set $\{\xi_j\}_{j=1}^J$ of independent and identically distributed random variables $\xi_1, \ldots, \xi_J \overset{iid}{\sim} \mathcal{P}_\xi$ is called a mini-batch. Inspired by [40], we use the following notation for the stochastic estimates computed from a mini-batch of size $J$:

$$\overline{\nabla}^J f(x) = \tfrac{1}{J} \sum_{j=1}^J \nabla f(x, \xi_j), \quad \overline{c}^J(x) = \tfrac{1}{J} \sum_{j=1}^J c(x, \zeta_j), \quad \overline{\nabla}^J c(x) = \tfrac{1}{J} \sum_{j=1}^J \nabla c(x, \zeta_j). \tag{6}$$

Table 2: Assumptions on objective and constraint functions, $F$ and $C$, which allow for theoretical convergence proofs.

| Algorithm | Objective function $F$ | | | | Constraint function $C$ | | | | | | |
| --- | --- | --- | --- | --- | --- | --- | --- | --- | --- | --- | --- |
| | stochastic | weakly convex | $\mathcal{C}^1$ with Lipschitz $\nabla F$ | tame loc. Lipschitz | stochastic | $C(x) = 0$ | $C(x) = 0$ and $C(x) \leq 0$ | linear | weakly convex | $\mathcal{C}^1$ with Lipschitz $\nabla C$ | tame loc. Lipschitz |
| SGD | ✓ | (✓) | (✓) | ✓ | | | | | | | |
| [6] [29] [17] | ✓ | – | ✓ | – | – | ✓ | – | – | – | ✓ | – |
| [40] | ✓ | – | ✓($\mathcal{C}^3$) | – | – | ✓ | – | – | – | ✓($\mathcal{C}^3$) | – |
| [49] [18] | ✓ | – | ✓ | – | – | (✓) | ✓ | – | – | ✓ | – |
| [41] | ✓ | – | ✓($\mathcal{C}^2$) | – | – | (✓) | ✓ | – | – | ✓($\mathcal{C}^2$) | – |
| [10] | ✓ | – | ✓(+ cvx) | – | – | ✓ | – | ✓ | – | – | – |
| [43] | ✓ | – | ✓ | – | ✓ | ✓ | – | – | – | ✓ | – |
| SSL-ALM [32] | ✓ | – | ✓ | – | ✓ | (✓) | ✓ | ✓ | – | – | – |
| Stoch. Ghost [27] | ✓ | – | ✓ | – | ✓ | (✓) | ✓ | – | – | ✓ | – |
| Stoch. Switch. Subg. [33] | ✓ | ✓ | – | – | ✓ | (✓) | ✓ | – | ✓ | – | – |

## 3.1 Review of methods for constrained ERM

We compare recent constrained optimization algorithms considering a stochastic objective function in Table 2. We note that most of them do not consider the case of stochastic constraints. Among those which do consider stochastic constraints, only three admit inequality constraints. Moreover, with the exception of [33], all the algorithms in Table 2 assume $F$ to be at least $\mathcal{C}^1$, which makes addressing the challenge of nonsmoothness of $F$ infeasible. The recent paper [19] leads us to the conclusion that assuming the objective and constraint functions to be tame and locally Lipschitz is a suitable requirement for solving (5) with theoretical guarantees of convergence. At this point, however, no such algorithm exists, to the best of our knowledge.

Consequently, we consider the practical performance of the algorithms that address the challenges of solving (5) most closely: Stoch. Ghost [27], SSL-ALM [32], and Stoch. Switching Subgradient [33].

## 3.2 Stochastic Ghost Method (StGh)

The Stochastic Ghost method was described in [27] where a method for solving (1) in the non-stochastic setting [28] was combined with the stochastic sampling inspired by an unbiased Monte Carlo method [9]. The method [28] for the non-stochastic setting is based on solving subproblem (7) to obtain a direction $d$ to preform the classical line search. Here, $e \in \mathbb{R}^m$ is a vector with all elements equal to one, $\tau$ and $\beta > 0$ are user-prescribed constants and $\kappa_k$ is defined as a certain convex combination of optimization subproblems related to $C$ and $\nabla C$. The definition of $\kappa_k$ enables to expand the feasibility region so that (7) is always feasible. As the problem (1) is stochastic, the subproblem (7) is modified to a stochastic version (8), using the notation in (6):

$$
\begin{aligned}
\min_d \quad & \nabla F(x_k)^\top d + \tfrac{\tau}{2}\|d\|^2, \\
\text{s.t.} \quad & C(x_k) + \nabla C(x_k)^\top d \leq \kappa_k e, \qquad (7) \\
& \|d\|_\infty \leq \beta,
\end{aligned}
\qquad
\begin{aligned}
\min_d \quad & \overline{\nabla}^J f(x_k)^\top d + \tfrac{\tau}{2}\|d\|^2, \\
\text{s.t.} \quad & \overline{c}^J(x_k) + \overline{\nabla}^J c(x_k)^\top d \leq \overline{\kappa_k}^J e, \qquad (8) \\
& \|d\|_\infty \leq \beta.
\end{aligned}
$$

In the stochastic setting, an unbiased estimate $d(x_k)$ of the line search direction $d$ is needed and it is computed using four particular mini-batches as follows. To facilitate comprehension, we denote $X_k^J = \{X_{k,j}\}_{j=1}^J$ a mini-batch of size $J$ with the $j$-th element $X_{k,j} = (\nabla f(x_k, \xi_{k,j}), c(x_k, \zeta_{k,j}), \nabla c(x_k, \zeta_{k,j}))$. First, we sample a random variable $N \sim \mathcal{G}(p_0)$ from the geometric distribution. Then we sample the mini-batches $X_k^1$ and $X_k^{2^{N+1}}$ and we partition the mini-batch $X_k^{2^{N+1}}$ of size $2^{N+1}$ into two mini-batches $\mathrm{odd}(X_k^{2^{N+1}})$ and $\mathrm{even}(X_k^{2^{N+1}})$ of size $2^N$. Finally, we solve (8) for each of the four mini-batches, denoting by $d(x_k; X_k^J)$ the solution of (8) for

the corresponding mini-batch $X_k^J$. We obtain

$$d(x_k) = \frac{d(x_k; X_k^{2^{N+1}}) - \frac{1}{2}\left(d(x_k; \mathrm{odd}(X_k^{2^{N+1}})) + d(x_k; \mathrm{even}(X_k^{2^{N+1}}))\right)}{(1 - p_0)^N p_0} + d(x_k; X_k^1). \quad (9)$$

An update between the iterations $x_k$ and $x_{k+1}$ is computed as

$$x_{k+1} = x_k + \alpha_k d(x_k),$$

where the deterministic stepsize $\alpha_k$ fulfills the classical requirement to be square-summable $\sum_{k=1}^{\infty} (\alpha_k)^2 < \infty$ but not summable $\sum_{k=1}^{\infty} \alpha_k = \infty$. For more details, see Algorithm 1.

### 3.3 Stochastic Smoothed and Linearized AL Method (SSL-ALM)

The Stochastic Smoothed and Linearized AL Method (SSL-ALM) was described in [32] for optimization problems with stochastic linear constraints. Although problem (1) has non-linear inequality constraints, we use the SSL-ALM due to the lack of algorithms in the literature dealing with stochastic non-linear constraints; see Table 2. The transition between equality and inequality constraints is handled with slack variables. Following the structure of [32], we minimize over the set $\mathcal{X} = \mathbb{R}^n \times \mathbb{R}_{\geq 0}^m$. The method is based on the augmented Lagrangian (AL) function $L_\rho(x, y) = F(x) + y^\top C(x) + \frac{\rho}{2}\|C(x)\|^2$, which is a result of merging the Lagrange function with the penalty methods [7]. Adding a smoothing term yields the proximal AL function

$$K_{\rho,\mu}(x, y, z) = L_\rho(x, y) + \frac{\mu}{2}\|x - z\|^2.$$

The SSL-ALM method was originally proposed in [32] where it is interpreted as an inexact gradient descent step on the Moreau envelope. An important property of the Moreau envelope is that its stationary points coincide with those of the original function.

The strength of this method is that, as opposed to the Stochastic Ghost method, it does not use large mini-batch sizes. In each iteration, we sample $\xi \overset{iid}{\sim} \mathcal{P}_\xi$ to evaluate the objective and $\zeta_1, \zeta_2 \overset{iid}{\sim} \mathcal{P}_\zeta$ to evaluate the constraint function and its Jacobian matrix, respectively. The function

$$G(x, y, z; \xi, \zeta_1, \zeta_2) = \nabla f(x, \xi) + \nabla c(x, \zeta_1)^\top y + \rho \nabla c(x, \zeta_1)^\top c(x, \zeta_2) + \mu(x - z) \quad (10)$$

is defined so that, in iteration $k$, $\mathbb{E}_{\xi,\zeta_1,\zeta_2}[G(x_k, y_{k+1}, z_k; \xi, \zeta_1, \zeta_2)] = \nabla K_{\rho,\mu}(x_k, y_{k+1}, z_k)$. Omitting some details, the updates are performed using some parameters $\eta, \tau$, and $\beta$ as follows:

$$\begin{aligned} y_{k+1} &= y_k + \eta c(x, \zeta_1), \\ x_{k+1} &= \mathrm{proj}_{\mathcal{X}}(x_k - \tau G(x_k, y_{k+1}, z_k; \xi, \zeta_1, \zeta_2)), \\ z_{k+1} &= z_k + \beta(x_k - z_k). \end{aligned} \quad (11)$$

For more details, see Algorithm 2.

### 3.4 Stochastic Switching Subgradient Method (SSw)

The Stochastic Switching Subgradient method was described in [33] for optimization problems over a closed convex set $\mathcal{X} \subset \mathbb{R}^d$ which is easy to project on and for weakly convex objective and constraint functions $F$ and $C$ which may be non-smooth. This is why the notion of gradient of $F$ and $C$ is replaced by a more general notion of subgradient, which is an element of a subdifferential.

The algorithm requires as input a prescribed sequence of infeasibility tolerances $\epsilon_k$ and sequences of stepsizes $\eta_k^f$ and $\eta_k^c$. In iteration $k$, we sample $\zeta_1, \ldots, \zeta_J \overset{iid}{\sim} \mathcal{P}_\zeta$ to compute an estimate $\overline{c}^J(x_k)$. If $\overline{c}^J(x_k)$ is smaller than $\epsilon_k$, we sample $\xi \overset{iid}{\sim} \mathcal{P}_\xi$ and an update between $x_k$ and $x_{k+1}$ is computed using a stochastic estimate $S^f(x_k, \xi)$ of an element of the subdifferential $\partial F(x_k)$ of the objective function:

$$x_{k+1} = \mathrm{proj}_{\mathcal{X}}(x_k - \eta_k^f S^f(x_k, \xi)).$$

Otherwise, we sample $\zeta \overset{iid}{\sim} \mathcal{P}_\zeta$ and the update is computed using a stochastic estimate $S^c(x_k, \zeta)$ of an element of the subdifferential $\partial C(x_k)$ of the constraint function:

$$x_{k+1} = \mathrm{proj}_{\mathcal{X}}(x_k - \eta_k^c S^c(x_k, \zeta)).$$

In either case, the updates are only saved starting from a prescribed index $k_0$ and the final output is sampled randomly from the saved updates. For more details, see Algorithm 3. The algorithm presented here is slightly more general than the one presented in [33]: we allow for the possibility of different stepsizes for the objective update, $\eta_k^f$, and the constraint update $\eta_k^c$, while the original method employs equal stepsizes $\eta_k^f = \eta_k^c$.

# 4 Experimental evaluation

In this section, we illustrate the presented algorithms on a real-world instance of the ACS dataset, comparing how they fare with optimization and fairness metrics.

## 4.1 Dataset for fair ML

[22] proposed a large-scale dataset for fair Machine Learning, based on the ACS PUMS data sample (American Community Survey Public Use Microdata Sample). The ACS survey is sent annually to approximately 3.5 million US households in order to gather information on features such as ancestry, citizenship, education, employment, or income. Therefore, it has the potential to give rise to large-scale learning and optimization problems.

We use the ACSIncome dataset over the state of Oklahoma, and choose the binary classification task of predicting whether an individual's income is over \$50,000. The dataset contains 9 features and 17,917 data points, and may be accessed via the Python package Folktables. We choose race (**RAC1P**) as the protected attribute. In the original dataset, it is a categorical variable with 9 values. For the purposes of this experiment, we binarize it to obtain the non-protected group of "white" people and the protected group of "non-white" people. The dataset is split randomly into train (80%, 14,333 points) and test (20%, 3,584 points) subsets and it is stratified with respect to the protected attribute, i.e., the proportion of "white" and "non-white" samples in the training and test sets is equivalent to that in the full dataset (30.8% of positive labels in group "white", 20.7% in the group "non-white"). The protected attributes are then removed from the data so that the model cannot learn from them directly. The data is normalized using Scikit-Learn StandardScaler.

Note that ACSIncome is a real-world dataset for which ERM-based predictors without fairness safeguards are known to learn biases [30]. Accordingly, Table 3 (line 1) shows that an ERM predictor without fairness safeguards has poor fairness metrics; see also Figure 4.

## 4.2 Experiments

**Numerical setup.** Experiments are conducted on an Asus Zenbook UX535 laptop with AMD Ryzen 7 5800H CPU, and 16GB RAM. The code is written in Python with the PyTorch package [44].

**Problems.** We consider the constrained ERM problem (4) – $\mathcal{R} = 0$, and, as baselines, the ERM problem (2) without any regularization, $\mathcal{R} = 0$, and with a fairness inducing regularizer $\mathcal{R}$ that promotes small difference in accuracy between groups, provided by the Fairret library [11]. In all problems, we take as loss function the Binary Cross Entropy with Logits Loss

$$\ell(f_\theta(X_i), Y_i) = -Y_i \cdot \log \sigma(f_\theta(X_i)) - (1 - Y_i) \cdot \log(1 - \sigma(f_\theta(X_i))), \tag{12}$$

where $\sigma(z) = \frac{1}{1+e^{-z}}$ is the sigmoid function, and the prediction function $f_\theta$ is a neural network with 2 interconnected hidden layers of sizes 64 and 32 and ReLU activation, with a total of 194 parameters.

**Algorithms and parameters.** We assess the performance of four algorithms for solving the constrained problem (4): *(1)* Stochastic Ghost (StGh) (Sec. 3.2 - parameters $p_0 = 0.4$, $\alpha_0 = 0.05$, $\rho = 0.8$, $\tau = 1$, $\beta = 10$, $\lambda = 0.5$, $\hat{\alpha} = 0.05$), *(2)* SSL-ALM (Sec. 3.3 - parameters $\mu = 2.0$, $\rho = 1.0$, $\tau = 0.01$, $\eta = 0.05$, $\beta = 0.5$, $M_y = 10$), *(3)* plain Augmented Lagrangian Method ALM (Sec. 3.3, smoothing term removed $\mu = 0$, otherwise the same setting as SSL-ALM), and *(4)* Stochastic Switching Subgradient (SSw) (Sec. 3.4 - $\eta_k^f = 0.5$, $\eta_k^c = 0.05$, $\epsilon_k = 10^{-4}$ if $k < 500$, $\epsilon_k = 0.97\epsilon_{k-1}$ for every $k \geq 500$ at each epoch). We also provide the behavior of SGD for solving the ERM problem, both with no fairness safeguards (SGD), and with fairness regularization provided by the Fairret library [11] (SGD-Fairret). These methods serve as baselines. When estimating the constraints, we sample an equal number of data points for every subgroup.

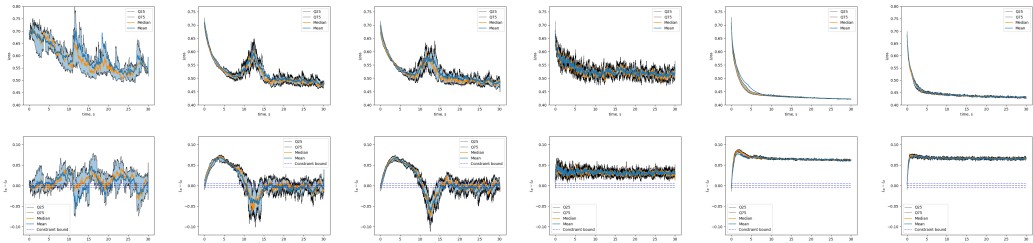

Figure 1: Train loss and constraint values (first and second row) over time (s) on the ACS Income dataset for each algorithm. From left to right: StGh, SSL-ALM, ALM, SSw, SGD, SGD-Fairret.

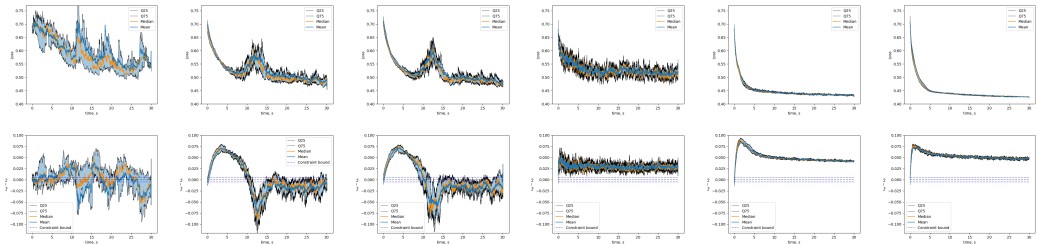

Figure 2: Test loss and constraint (first and second row) values over time (s) on the ACS Income dataset for each algorithm. From left to right: StGh, SSL-ALM, ALM, SSw, SGD, SGD-Fairret.

**Optimization performance.** Figures 1 and 2 present the evolution of loss and constraint values over the train and test datasets for the four algorithms addressing the constrained problem (columns 1–4), as well as for the two baselines: SGD without fairness (col. 5), and SGD with fairness regularization (col. 6). Each algorithm is run 10 times, and the plots display the mean, median, and quartiles values.

To a certain extent, the four algorithms (col. 1–4) succeed in minimizing the loss and satisfying the constraints on the train set. The AL-based methods (col. 2 and 3) demonstrate a better behavior compared to StGh and SSw; indeed, StGh exhibits higher variability in both loss and constraint values (col. 1), while SSw fails to satisfy the constraints within the required bounds (col. 4). We were unable to identify parameter settings for SSw that simultaneously satisfy the constraints and minimize the objective function. Appendix B provides the behavior of SSw with equal objective and constraint and stepsizes; the constraints are satisfied well, but the objective function is barely minimized. The ERM baselines (col. 5 and 6) exhibit lower variability in the trajectories, and minimize the loss in less time, but as expected, they do not satisfy the constraints.

The ALM and SSL-ALM schemes are the closest to satisfying the constraints on the train set. On the test set, however, they are slightly biased towards negative values. Such bias is expected on unseen data and reflects the generalization behavior of fairness-constrained estimators. This is beyond the scope of the current work; see e.g. [12].

**Fairness performance.** Figure 3 presents the distribution of predictions over both groups. The distribution of prediction without fairness guarantees (col. 5) clearly does not meet the group fairness standard. Indeed, the "non-white" group has a significantly higher likelihood than the "white" group of receiving small predicted values, and the converse holds for large predicted values. The SGD-Fairret model (col. 6) lies between the four constrained models and SGD. Among the fairness-constrained models, the ALM and SSL-ALM distributions are the closest to the distributions of SGD without fairness, which is consistent with retaining good prediction information. The four models that approximately solve the fairness formulation (col. 1–4) all have closer distributions across groups. Numerically, this is expressed in Table 3 (col. Wd), which reports the value of the Wasserstein distance between group distributions for each model.

Table 3 displays the fairness metrics presented in Section 2: independence (Ind), separation (Sp), and sufficiency (Sf), along with inaccuracy (Ina). The mean value and standard deviation over 10 runs are presented for the four fairness-constrained models and the two baselines, both on train and test sets. Figure 4 presents the mean values as spider plots. For all metrics, smaller is better.

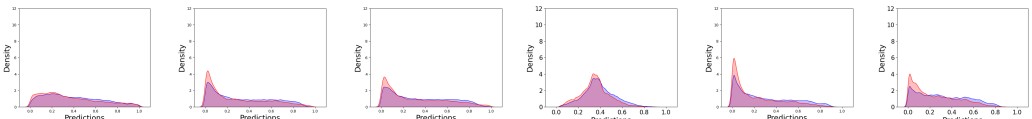

Figure 3: Distribution of predictions for each algorithm. Left to right: StGh, SSL-ALM, ALM, SSw, SGD, SGD-Fairret. Blue and red denote "white" and "non-white" groups.

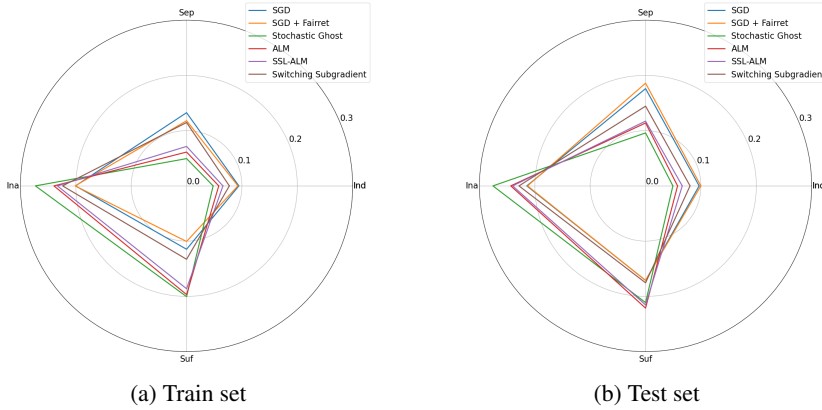

(a) Train set        (b) Test set

Figure 4: Average value of the three fairness metrics (independence (Ind), separation (Sp), and sufficiency (Sf)), along with inaccuracy (Ina). For all metrics, smaller values are better.

Table 3: Fairness metrics (independence, separation, sufficiency), inaccuracy, and Wasserstein distances between groups (Wd) for the four constrained estimators and the two baselines.

| Algname | Train | | | | | Test | | | | |
|---|---|---|---|---|---|---|---|---|---|---|
| | Ind | Sp | Ina | Sf | Wd | Ind | Sp | Ina | Sf | Wd |
| SGD | $0{,}094_{\pm0{,}004}$ | $0{,}132_{\pm0{,}007}$ | $\mathbf{0{,}201}_{\pm0{,}001}$ | $0{,}115_{\pm0{,}006}$ | $0{,}008_{\pm0{,}000}$ | $0{,}097_{\pm0{,}006}$ | $0{,}176_{\pm0{,}016}$ | $0{,}215_{\pm0{,}002}$ | $\mathbf{0{,}171}_{\pm0{,}009}$ | $0{,}008_{\pm0{,}000}$ |
| StGh | $\mathbf{0{,}048}_{\pm0{,}026}$ | $\mathbf{0{,}049}_{\pm0{,}028}$ | $0{,}273_{\pm0{,}024}$ | $0{,}200_{\pm0{,}038}$ | $0{,}002_{\pm0{,}001}$ | $\mathbf{0{,}049}_{\pm0{,}029}$ | $\mathbf{0{,}096}_{\pm0{,}039}$ | $0{,}276_{\pm0{,}022}$ | $0{,}211_{\pm0{,}033}$ | $0{,}003_{\pm0{,}002}$ |
| ALM | $0{,}058_{\pm0{,}007}$ | $0{,}061_{\pm0{,}016}$ | $0{,}240_{\pm0{,}012}$ | $0{,}197_{\pm0{,}011}$ | $0{,}003_{\pm0{,}000}$ | $0{,}058_{\pm0{,}012}$ | $0{,}114_{\pm0{,}014}$ | $0{,}244_{\pm0{,}007}$ | $0{,}221_{\pm0{,}017}$ | $0{,}003_{\pm0{,}001}$ |
| SSL-ALM | $0{,}066_{\pm0{,}009}$ | $0{,}071_{\pm0{,}015}$ | $0{,}233_{\pm0{,}017}$ | $0{,}186_{\pm0{,}013}$ | $0{,}003_{\pm0{,}001}$ | $0{,}066_{\pm0{,}011}$ | $0{,}117_{\pm0{,}023}$ | $0{,}240_{\pm0{,}012}$ | $0{,}215_{\pm0{,}022}$ | $0{,}004_{\pm0{,}001}$ |
| SSw | $0{,}077_{\pm0{,}029}$ | $0{,}115_{\pm0{,}029}$ | $0{,}224_{\pm0{,}017}$ | $0{,}133_{\pm0{,}015}$ | $\mathbf{0{,}001}_{\pm0{,}001}$ | $0{,}080_{\pm0{,}029}$ | $0{,}144_{\pm0{,}050}$ | $0{,}229_{\pm0{,}013}$ | $0{,}175_{\pm0{,}031}$ | $\mathbf{0{,}002}_{\pm0{,}001}$ |
| SGD-Fairret | $0{,}091_{\pm0{,}012}$ | $0{,}121_{\pm0{,}017}$ | $\mathbf{0{,}201}_{\pm0{,}002}$ | $\mathbf{0{,}106}_{\pm0{,}010}$ | $0{,}005_{\pm0{,}001}$ | $0{,}094_{\pm0{,}010}$ | $0{,}174_{\pm0{,}019}$ | $\mathbf{0{,}213}_{\pm0{,}002}$ | $0{,}180_{\pm0{,}022}$ | $0{,}006_{\pm0{,}001}$ |

Among the four fairness-constrained models, StGh performs best in terms of independence and separation, but worst in terms of accuracy. SSw achieves fairness and accuracy metrics that have intermediate values relative to those of the unconstrained SGD model, and those of the other constrained models. This is consistent with the observation that the optimization method, with our choice of parameters, favored minimizing the objective over satisfying the constraints. The ALM and SSL-ALM methods provide the best compromise: they improve independence and separation relative to the SGD model, while moderately degrading accuracy. SGD-Fairret slightly improves sufficiency relative to the SGD model. The four models constrained in the difference of loss between subgroups have higher values of sufficiency. Similar observations hold for metrics on the test set.

## 5 Conclusion

To the best of our knowledge, this paper provides the first benchmark for assessing the performance of optimization methods on real-world instances of fairness constrained training of models. We highlight the challenges of this approach, namely that objective and constraints are non-convex, non-smooth, and large-scale, and review the performance of four practical algorithms.

**Limitations** Our work identifies that there is currently no algorithm with guarantees for solving the fairness constrained problem. Above all, we hope that this work, along with the Python toolbox for easy benchmarking of new optimization methods, will stimulate further interest in this topic. Also, we caution readers that the method present here is not a silver-bullet that handles all biases and ethical issues of training ML models. In particular, care must be taken that fair ML is part of a interdisciplinary pipeline that integrates the specifics of the use-case, and that it does not serve as an excuse for pursuing Business-As-Usual policies that fail to tackle ethical issues [2, 52].

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

# A Algorithms in more detail

In this section, we provide the pseudocodes of algorithms presented in Section 3 as Algorithms 1 to 3. Recall that we denote by $X_k^J = \{X_{k,j}\}_{j=1}^J$ a mini-batch of size $J$ with the $j$-th element

$$X_{k,j} = (\nabla f(x_k, \xi_{k,j}), c(x_k, \zeta_{k,j}), \nabla c(x_k, \zeta_{k,j})). \tag{13}$$

# B Additional experiments on SSw

This Section provides additional information on the behavior of SSw. Figure 5 shows the evolution of the objective value and constraints for 10 runs of the SSw algorithm, over train and test, with equal objective and constraint stepsizes $\eta_k^f = \eta_k^c = 0.02$. In that case, the constraints are satisfied well, but the objective function is barely minimized.

---

**Algorithm 1** Stochastic Ghost algorithm

---

**Require:** Training dataset $\mathcal{D}$, constraint dataset $\mathcal{C}$, initial neural network weights $x_0$
**Require:** Parameters $p_0 \in (0,1)$, $\alpha_0$, $\hat{\alpha}$, $\rho$, $\tau$, $\beta$
 1: **for** Iteration $k = 0$ **to** $K - 1$ **do**
 2:     Sample $\xi \overset{iid}{\sim} \mathcal{P}_\xi$ and $\zeta \overset{iid}{\sim} \mathcal{P}_\zeta$
 3:     Sample $N \sim \mathcal{G}(p_0)$
 4:     Set $J = 2^{N+1}$
 5:     Sample a mini-batch $\{\zeta_j\}_{j=1}^J$ so that $\zeta_1, \ldots, \zeta_J \overset{iid}{\sim} \mathcal{P}_\zeta$
 6:     Sample a mini-batch $\{\xi_j\}_{j=1}^J$ so that $\xi_1, \ldots, \xi_J \overset{iid}{\sim} \mathcal{P}_\xi$
 7:     Set $X_k^1$ and $X_k^{2^{N+1}}$ using (13)
 8:     Compute $d(x_k)$ from (9)
 9:     Set $\alpha_k = \alpha_{k-1}(1 - \hat{\alpha}\alpha_{k-1})$
10:     Update $x_{k+1} = x_k + \alpha_k d(x_k)$
11: **end for**

---

---

**Algorithm 2** Stochastic Smoothed and Linearized AL Method for solving (1)

---

**Require:** Training dataset $\mathcal{D}$, constraint dataset $\mathcal{C}$, initial neural network weights $x_0$
**Require:** Parameters $\mu$, $\eta$, $M_y > 0$, $\tau$, $\beta$, $\rho \geq 0$
 1: **for** Iteration $k = 0$ **to** $K - 1$ **do**
 2:     Sample $\xi \overset{iid}{\sim} \mathcal{P}_\xi$ and $\zeta_1, \zeta_2 \overset{iid}{\sim} \mathcal{P}_\zeta$
 3:     $y_{k+1} = y_k + \eta c(x, \zeta_1)$
 4:     **if** $||y_{k+1}|| \geq M_y$ **then**
 5:         $y_{k+1} = 0$
 6:     **end if**
 7:     $x_{k+1} = \text{proj}_\mathcal{X}(x_k - \tau G(x_k, y_{k+1}, z_k; \xi, \zeta_1, \zeta_2))$, where $G$ is defined in (10)
 8:     $z_{k+1} = z_k + \beta(x_k - z_k)$
 9: **end for**

---

---

**Algorithm 3** Stochastic Switching Subgradient Method

---

**Require:** Training dataset $\mathcal{D}$, constraint dataset $\mathcal{C}$, initial neural network weights $x_0 \in \mathcal{X}$
**Require:** Total number of iterations $K$, sequence of tolerances of infeasibility $\epsilon_k \geq 0$, sequences of
     stepsizes $\eta_k^f$ and $\eta_k^c$, mini-batch size $J$, starting index $k_0$ for recording outputs, $I = \emptyset$
 1: **for** Iteration $k = 0$ **to** $K - 1$ **do**
 2:     Sample a mini-batch $\{\zeta_j\}_{j=1}^J$ so that $\zeta_1, \ldots, \zeta_J \overset{iid}{\sim} \mathcal{P}_\zeta$
 3:     Set $\bar{c}^J(x_k) = \frac{1}{J} \sum_{j=1}^J c(x_k, \zeta_j)$
 4:     **if** $\bar{c}^J(x_k) \leq \epsilon_k$ **then**
 5:         Sample $\xi \overset{iid}{\sim} \mathcal{P}_\xi$ and generate $S^f(x_k, \xi)$
 6:         Set $x_{k+1} = \text{proj}_\mathcal{X}(x_k - \eta_k^f S^f(x_k, \xi))$ and, if $k \geq k_0$, $I = I \cup \{k\}$
 7:     **else**
 8:         Sample $\zeta \overset{iid}{\sim} \mathcal{P}_\zeta$ and generate $S^c(x_k, \zeta)$
 9:         Set $x_{k+1} = \text{proj}_\mathcal{X}(x_k - \eta_k^c S^c(x_k, \zeta))$ and, if $k \geq k_0$, $I = I \cup \{k\}$
10:     **end if**
11: **end for**
12: **Output:** $x_\tau$ with $\tau$ randomly sampled from $I$ using $P(\tau = k) = \frac{\eta_k}{\sum_{s \in I} \eta_s}$.

---

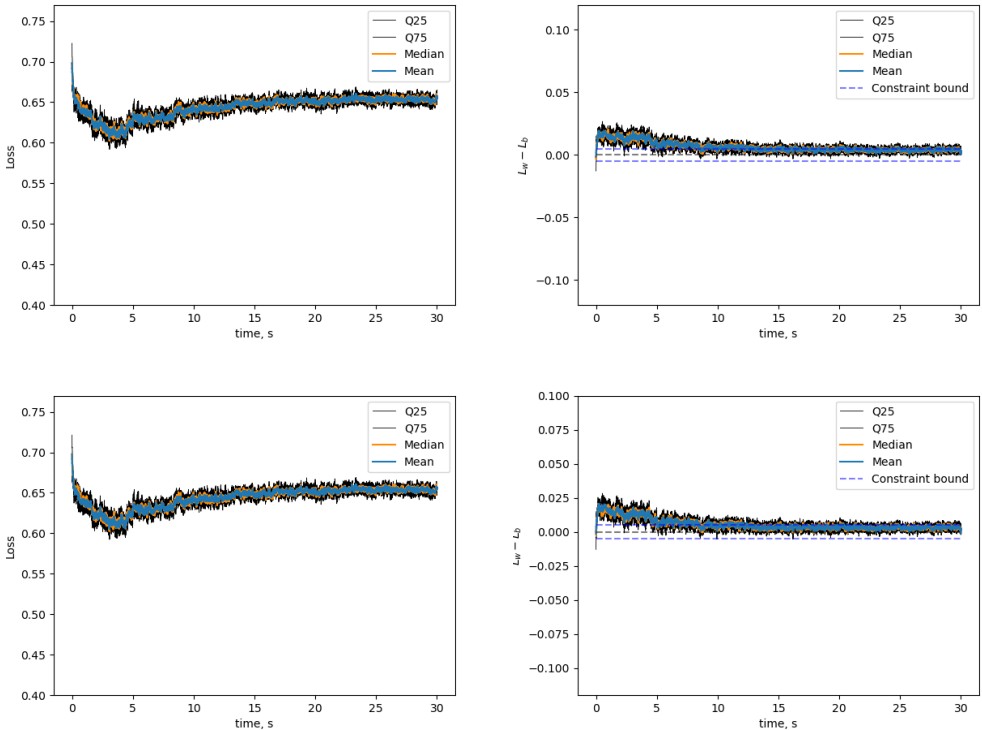

Figure 5: Loss and constraint values over time (s) on the train and test set (first and second row) on the ACS Income dataset for the SSw algorithm.

