# OpenReview forum: "Benchmarking Stochastic Approximation Algorithms for Fairness-Constrained Training of Deep Neural Networks"
_NeurIPS.cc/2025/Datasets_and_Benchmarks_Track — Submitted to NeurIPS 2025 Datasets and Benchmarks Track_

### Official Review · Reviewer_kL8E · 2025-06-11

**Rating:** 5
**Confidence:** 3

**Summary:**

This paper introduces a benchmark for the fairness-constraint training scenario of deep neural networks. The challenges of DNN training under fairness-constraints are reviewed, and to evaluate the ability of training algorithms to deal with the challenges, several algorithms, such as StGh, SSL-ALM, and SSw, are included in the benchmark as baselines. To show the advantages of the proposed benchmark, the authors tested three unimplemented algorithms, and considered the optimization performance and improvement by using the benchmark.

**Dataset Code Accessibility:**

Yes

**Dataset Code Comments:**

The github link to the benchmark is included in the paper (https://github.com/humancompatible/train, see page 1 of the paper). The github page includes the codes of the benchmark, as well as the installation and reproducing  guides. Even though the data has not directly uploaded to the github page, a python code to download and setup data is provided.

**Ethical Considerations:**

No, there are no or only very minor ethics concerns

**Final Justification:**

The authors' feedback solves most of my concerns. Thus I support that this paper can be accepted.

**Limitations Weaknesses:**

1. The paper mainly focus on the baseline methods in the benchmark. It is better to include more details of the data in the benchmark.

2. The experimental results only include the performance of the baseline methods. It is better to include the test performance of more third-party methods to show the generalization ability of the benchmark.

**Strengths Contributions:**

1. The paper considers the fairness-constraint training problem of DNNs, and designs a benchmark to measure the performance of learning algorithms.

2. The learning problem is well-described, and several baseline training algorithms are reviewed and included into the benchmark.

3. The three learing algorithms are described mathmetically, which provide enough theoretical basis to support the design of the benchmark.

---

> ### Author Rebuttal · Authors · 2025-07-31
>
> We thank you for your careful reading and comments. Here are further details.
>
> > The paper mainly focus on the baseline methods in the benchmark. It is better to include more details of the data in the benchmark.
>
> We stress that our contribution is the theoretical formulation of constraints (eq. 4) and the toolbox that implements three yet-unimplemented algorithms and provides an easy-to-use benchmark. We employ data from the folktables toolbox, that corresponds to [22]. We describe this and how we use that data in Section 4.1. We do not claim that the dataset is part of our contributions. Besides, our toolbox allows to easily switch the folktables dataset with any other dataset. We feel that we may be misunderstanding your comment. If so, could you please rephrase your demand? We will be happy to answer.
>
> > The experimental results only include the performance of the baseline methods. It is better to include the test performance of more third-party methods to show the generalization ability of the benchmark.
>
> As we discuss in Section 2, fairness is enforced on AI models by either pre-processing the dataset, changing the training of the method (in-processing), or adding safeguards to a classical model that was trained with fairness components (post-processing). As our setting falls into the scope of in-processing, comparison with third-party methods only makes sense for other in-processing methods. The Fairret toolbox is, to the best of our knowledge, the only in-processing method with available and easy-to-use implementation, and we do compare against it. Moreover, we would like to stress that we have reviewed all the algorithms that may be applicable to training deep neural networks with fairness constraints, and we provided implementation of the most promising ones.
>
> We hope to have addressed all your concerns. We remain at your disposal may you have any further questions or require additional information. We would be grateful if you could consider revising your score based on the answers we provided.

---

### Official Review · Reviewer_1P8v · 2025-07-02

**Rating:** 3
**Confidence:** 3

**Summary:**

In this paper, the authors study the performance of three stochastic approximation algorithms for fairness-constrained training of DNNs. The idea is that many fairness problems in machine learning can be written as constrained empirical risk minimization (ERM), where we try to minimize the loss but also keep fairness constraints satisfied across different groups. The authors review existing algorithms for solving such ERM problems, implement four existing algorithms that are applicable in practice, and evaluate them on one real-world dataset. They also compare these algorithms with alternative fairness-aware approaches and standard unconstrained baselines. Specifically, the algorithms that they implemented are:

- Stochastic Ghost (StGh)
- Stochastic Smoothed and Linearized ALM (SSL-ALM)
- Standard ALM (variant of SSL-ALM without smoothing)
- Stochastic Switching Subgradient (SSw)

**Dataset Code Accessibility:**

Yes

**Dataset Code Comments:**

The code is available on GitHub with clear instructions, everything needed to reproduce the experiments, including setup and parameter choices is provided. it is straightforward to run and verify the results.

**Ethical Considerations:**

No, there are no or only very minor ethics concerns

**Final Justification:**

Overall, the paper makes a valuable contribution, but its scope feels somewhat narrow. As noted in my rebuttal, the evaluation remains limited, and the work would greatly benefit from even a light ablation or basic robustness check, which could significantly strengthen the practical insights given the paper’s empirical focus.

**Limitations Weaknesses:**

Although I do appreciate the relevance and importance of this work, I have two major concerns/comments:

- I think the paper would benefit from a few more experiments on other datasets. Evaluating all the methods on just one dataset feels a bit limited, and it’s hard to tell if the results would generalize.
- Analyzing the sensitivity of key hyper parameters (e.g., mini-batch size, step sizes, etc.) would also improve the quality of this paper. It would give a better sense of how stable or fragile the results are, in terms of fairness and accuracy.

One more suggestion: In the experiments, the authors binarized the dataset into white and non-white. Is it possible to evaluate the full (or part of) the multi-class version? If that is not applicable, can you please explain why?

**Strengths Contributions:**

- This work addresses an important and timely topic with clear real-life relevance and motivation.
- The implementation of four existing algorithms within a unified framework is definitely valuable and can help benchmark future work on fairness-constrained optimization.
- The paper is generally well-written and easy to follow.
- This work reports results across multiple fairness metrics, which gives a more complete view of model behavior.

---

> ### Author Rebuttal · Authors · 2025-07-31
>
> We thank you for your careful reading and comments. Here are further details.
>
> > I think the paper would benefit from a few more experiments on other datasets. Evaluating all the methods on just one dataset feels a bit limited, and it’s hard to tell if the results would generalize. Analyzing the sensitivity of key hyper parameters (e.g., mini-batch size, step sizes, etc.) would also improve the quality of this paper. It would give a better sense of how stable or fragile the results are, in terms of fairness and accuracy.
>
> While we agree that working on one dataset provides limited insight in the generalization ability of the method, let us stress that ACSIncome “is a real-world dataset for which ERM-based predictors without fairness safeguards are known to learn biases [30]”, so that this is a realistic use-case. Besides, we believe that the main contributions of this work are the formulation of fair training of Deep Neural Networks as a constrained optimization problem, and the toolbox for benchmarking stochastic algorithms on such learning tasks. We believe that the current experimental section reasonably supports these two contributions, showing that ALM and Stochastic Ghost provide approximations of the solutions of (eq. 4). Whether these algorithms generalize well, and have reasonable hyperparameter tuning are difficult questions, beyond the scope of this work. We report on an experiment with a protected attribute taking $5$ values at the end of this rebuttal.
>
> > One more suggestion: In the experiments, the authors binarized the dataset into white and non-white. Is it possible to evaluate the full (or part of) the multi-class version? If that is not applicable, can you please explain why?
>
> Indeed, in the paper we chose to work with one binary protected attribute. It is possible and desirable to extend the formulation to attributes that have more than two values. However, a multi-class protected attribute with $n$ classes would require $2^n(2^n-1)/2$ constraints (the binomial coefficient 2 out of $2^n$), exponentially increasing the number of constraints, and thus the computational cost.
>
>
> ## Additional experiment
>
> We propose below two tables that describe an additional experiment. We consider the same dataset and learning task, with the marriage status (MAR) as protected attribute, that features 5 values. This is modeled by adding one constraint for each pair of values of the protected attribute, resulting in 10 constraints. The algorithms are run with a 3 minutes time budget, and the same hyperparameters as in the paper. Both tables provide the mean / quantile 0.25 / quantile 0.75 of corresponding values for the output model of each algorithm, over 10 runs, on the training set. In both tables, the lowest values in each line are shown in bold. Multiple values are in bold if they are very close to each other.
>
> The first table shows the constraints in the absolute value. We recall that constraints are satisfied when the absolute value is lower than 0.005. In that scenario, only SSw meets the constraints on average. ALM and SSL-ALM are within 180% of the constraint bound.
> | Algorithm | ALM | SGD | SSL-ALM | SSw | StGh | SGD-Fairret |
> | ---- | :----: | :----: | :----: | :----: | :----: | :----: |
> | mean | 0.0131 | 0.1267 | 0.0091 | **0.0016** | 0.0264 | 0.0936 |
> | quantile 0.25 | 	0.0051 | 0.0505 | 0.0032 | **0.0006** | 0.0075 | 0.0344 |
> | quantile 0.75 | 0.0186 | 0.2006 | 0.0128 | **0.0020** | 0.0285 | 0.1736 |
>
> The second table shows the fairness metrics (independence (Ind), separation (Sp), and sufficiency (Sf)), along with inaccuracy (Ina).
> | Algorithm | ALM | SGD | SSL-ALM | SSw | StGh | SGD-Fairret |
> | ---- | :----: | :----: | :----: | :----: | :----: | :----: |
> | Ina: mean | 0.2725 | **0.1649** | 0.2651 | 0.4515 | 0.3391 | 0.1724 |
> | Ina: q25 | 0.2641 | **0.1159** | 0.2604 | 0.3953 | 0.2838 | 0.1259 |
> | Ina: q75 | 0.2794 | **0.2084** | 0.2720 | 0.4915 | 0.3840 | 0.2136 |
> | Ind: mean | **0.0551** | 0.1136 | **0.0561** | 0.0865 | 0.0983 | 0.1211 |
> | Ind: q25 | **0.0233** | 0.0663 | **0.0266** | 0.0319 | 0.0386 | 0.0561 |
> | Ind: q75 | **0.0717** | 0.1572 | 0.0807 | 0.1037 | 0.1089 | 0.1624 |
> | Sf: mean | 0.3330 | **0.1428** | 0.3247 | 0.2615 | 0.2933 | 0.1859 |
> | Sf: q25 | 0.1837 | **0.0830** | 0.1724 | 0.1523 | 0.1645 | **0.0860** |
> | Sf: q75 | 0.4703 | **0.1971** | 0.4549 | 0.3681 | 0.4230 | 0.2837 |
> | Sp: mean | 0.1456 | **0.1309** | 0.1341 | 0.1762 | 0.2069 | 0.1574 |
> | Sp: q25 | 0.1055 | **0.0662** | 0.0872 | 0.0975 | 0.1060 | **0.0684** |
> | Sp: q75 | 0.1945 | 0.2110 | **0.1691** | 0.2195 | 0.2727 | 0.2414 |
>
> Overall, the results with the multiple-value attribute confirm the observations made for the binary attribute:
> - Concerning constraint satisfaction, Fairret regularization is better compared to SGD, but is difficult to use to reach a prescribed accuracy. SSw is the method that best meets the constraints, followed closely by SSL-ALM and ALM. Then, Stochastic Ghost ends with twice the constraint value of ALM on average.
> -  Concerning (in)accuracy, SGD performs best (0.16), as expected, and Fairret regularization follows closely (0.17). ALM and SSL-ALM have inaccuracy of (0.27). Then, Stochastic Ghost and SSw have inaccuracies of 0.34 and 0.45. In all of our experiments, these two methods perform poorly in terms of accuracy.
> - Concerning Independence, ALM and SSL-ALM perform best, as for the binary attribute case (see Table 3 in our paper). Fairret has performance comparable to SGD.
> - Concerning Sufficiency, as in the binary attribute case, ALM and SSL-ALM have values twice as large as SGD and Fairret,
> - Concerning Separation, SSL-ALM and ALM have values (0.13 / 0.15) comprised between SGD (0.13) and Fairret (0.16). In this case, these methods have similar performance, whereas for the binary attribute case, SSL-ALM and ALM had values twice as small as SGD and Fairret.
>
> We hope to have addressed all your concerns. We remain at your disposal may you have any further questions or require additional information. We would be grateful if you could consider revising your score based on the answers we provided.

---

> > ### Comment · Reviewer_1P8v · 2025-08-05
> >
> > Thank you for the detailed response. I truly appreciate the authors clarification and the additional experiment. This slightly strengthens the paper, and I acknowledge the effort to explore a higher-dimensional constraint setup.
> >
> > That said, I have decided to keep my original score, as the main concerns remain partially addressed.
> >
> > - Limited evaulation: While I do agree that Acsincome is a reasonable dataset, the entire experimental evaluation still hinges on a single dataset. Even the additional experiment uses the same dataset, which limits the generalizability of the findings.
> >
> > - Sensitivity analysis: "difficult questions, beyond the scope of this work". While this is understandable, even a light ablation or basic robustness check would significantly improve the practical insights, especially given the paper's empirical nature.
> >
> > Overall, the paper makes a useful contribution; however, the scope feels a bit narrow, in my opinion. I would like to thank the authors again for their response and for making their work accessible and reproducible.

---

> > > ### Author Response · Authors · 2025-08-08
> > > **Answer, and thank you for reviewing**
> > >
> > > We understand the strengths and concerns that you outlined in the review. We have done our best to provide an interesting contribution. We take note of your points about limited evaluation and sensitivity analysis for coming updates to our benchmark.
> > >
> > > We would like to thank the reviewer for time and effort invested to provide thorough and constructive feedback.
> > >
> > > Best regards, Authors

---

### Official Review · Reviewer_2yzk · 2025-07-02

**Rating:** 5
**Confidence:** 2

**Summary:**

In this work, the authors present a comprehensive study of fairness-constrained learning by applying three recently proposed approaches. Specifically, the paper includes a literature review of various methods for promoting fairness in deep neural networks (DNNs), formalizes a constraint function to enforce fairness, and introduces the associated evaluation metrics. The authors implement three different algorithms and evaluate them on the formulated optimization problem. Finally, the paper reports experimental results that assess both the performance of the applied algorithms and the fairness metrics using the ACSIncome dataset.

**Dataset Code Accessibility:**

Yes

**Ethical Considerations:**

No, there are no or only very minor ethics concerns

**Final Justification:**

The authors introduce additional experimental results for multi-valued attributes that confirms the observations made for the binary attributes.

Additionally, the authors extensively discussed the limitation of the proposed benchmark on handling a single attribute at time. Introducing such discussion in paper will further benefit future research direction.

The authors provide a valuable feedback addressing the other concerns.

**Limitations Weaknesses:**

My main concern is that the proposed method is limited to binarized features, meaning that it involves only two groups. Additionally, the problem is formulated as a single constraint problem, targeting one protected attribute. However, more realistic problems involve multi-valued categorical attributes  (e.g. also the protected attribute used, it is first binarized), as well as more than one protected attribution. Taking account the fact that the proposed method is evaluated on a single binary classification task, I want to ask the authors if the proposed benchmark can be easily generalized in a more realistic dataset, and what the relative computation complexity needs to be paid for such a generalization.

How the hyperparameters of each algorithm are chosen? Additionally, a study on correlation between evaluation metrics will be interesting. Finally, the experimental evaluation provided in this paper is quite limited.

Minor: The quality and readability of the figures need to be improved.

**Strengths Contributions:**

The paper is very well written and it deals with the highly relevant topic of fairness and bias in DNNs. Additionally, the paper provides a way of formalizing the non-convex empirical-risk problem, allowing to perform fairness benchmarks.

With the proposed formalization the papers unlock the capability of the evaluated algorithms to be applied in DNN setting.

---

> ### Author Rebuttal · Authors · 2025-07-31
>
> We thank you for your careful reading and comments. Here are further details.
>
> > My main concern is that the proposed method is limited to binarized features [...]. Additionally, the problem is formulated as a single constraint problem, targeting one protected attribute. However, more realistic problems involve multi-valued categorical attributes (e.g. also the protected attribute used, it is first binarized), as well as more than one protected attribution. Taking account the fact that the proposed method is evaluated on a single binary classification task, I want to ask the authors if the proposed benchmark can be easily generalized in a more realistic dataset, and what the relative computation complexity needs to be paid for such a generalization.
>
> Indeed, in the paper we chose to work with one binary protected attribute. It is of course possible and desirable to extend the formulation to attributes that have more than two values, and to several protected attributes. However, this is not straightforward:
> - for handling one attribute with $K$ values, $K \ge 2$, we can add a constraint for each pair of subgroups produced by the attribute, resulting in $K(K-1)/2$ constraints. We report such an experiment at the end of this rebuttal.
> - for handling several attributes at the same time, at least two options are possible.
>
> The first one consists in treating each attribute separately. This results in $n$ constraints, when $n$ binary attributes are considered. However, this approach has been shown to disregard intersectional biases, as discussed in [A Survey on Intersectional Fairness in Machine Learning: Notions, Mitigation, and Challenges, Gohar & Cheng 2023] and references therein.
>
> Another option is to split the population into small subgroups, according to the values of the joint protected attributes, and then add pairwise fairness comparisons to the problem. This results in (at most) $2^n(2^n-1)/2$ constraints (the binomial coefficient 2 out of $2^n$), when $n$ binary attributes are considered.
>
> We believe that this is an important and natural question to investigate. Nevertheless, the main motivation of this paper lies in investigating the possibility to solve optimization problems that feature large-scale constraints. Thus, we feel that this investigation lies outside the scope of the paper.
>
> Regarding the computation complexity, as discussed above, adding more values for the protected attribute, or more protected attributes translates by adding constraints to (eq. 4). The iteration complexity of the methods is the sum of (1) the number of samples required at each iteration, and (2) the cost of computing the updates. For instance, for SSL-ALM, $1 + 2C$ (times the batch size) samples are required at each iteration, where $C$ is the number of scalar constraints, and the updates have negligible costs. For Stochastic Ghost, the number of samples is larger (and random) at each iteration, and the updates are computed by solving Quadratic Programs. Besides, the computation complexity is given by the iteration complexity multiplied by the number of iterations required to reach a prescribed accuracy. We are not aware of estimates of that second quantity for each of the algorithms proposed.
>
> > How the hyperparameters of each algorithm are chosen?
>
> Regarding the hyperparameters, we have used the values suggested in the original papers for Stochastic Ghost, and SSL-ALM. In particular, we have not optimized these parameters on the learning task at hand. The performance of Switching Subgradient heavily depends on the constraint tolerance hyperparameter, and we have searched for a value that strikes a balance between optimizing loss and meeting constraints. That value proved elusive, which is a first indication that SSw does not perform as well as SSL-ALM or Stochastic Ghost.
>
> > Additionally, a study on correlation between evaluation metrics will be interesting.
>
> Fairness can be evaluated using many metrics. We chose the four suggested by the influential work [3]. A correlation study would indeed be interesting, but we believe that this belongs to the field of fairness and thus falls outside the scope of this work.
>
> >  Finally, the experimental evaluation provided in this paper is quite limited.
>
> While we agree that more experiments would be interesting fairness-wise, we stress that the main contribution of the paper is the fairness formulation for training Deep Neural Networks, and the toolbox for benchmarking stochastic algorithms on such learning tasks. We believe that the current experimental section reasonably supports these two contributions.
>
> > Minor: The quality and readability of the figures need to be improved.
>
> We will be happy to update the plots to vector graphic format, and improve readability (better scaling of figures 3 and 4).
>
> ## Additional experiment
>
> We propose below two tables that describe an additional experiment. We consider the same dataset and learning task, with the marriage status (MAR) as protected attribute, that features 5 values. This is modeled by adding one constraint for each pair of values of the protected attribute, resulting in 10 constraints. The algorithms are run with a 3 minutes time budget, and the same hyperparameters as in the paper. Both tables provide the mean / quantile 0.25 / quantile 0.75 of corresponding values for the output model of each algorithm, over 10 runs, on the training set. In both tables, the lowest values in each line are shown in bold. Multiple values are in bold if they are very close to each other.
>
> The first table shows the constraints in the absolute value. We recall that constraints are satisfied when the absolute value is lower than 0.005. In that scenario, only SSw meets the constraints on average. ALM and SSL-ALM are within 180% of the constraint bound.
> | Algorithm | ALM | SGD | SSL-ALM | SSw | StGh | SGD-Fairret |
> | ---- | :----: | :----: | :----: | :----: | :----: | :----: |
> | mean | 0.0131 | 0.1267 | 0.0091 | **0.0016** | 0.0264 | 0.0936 |
> | quantile 0.25 | 	0.0051 | 0.0505 | 0.0032 | **0.0006** | 0.0075 | 0.0344 |
> | quantile 0.75 | 0.0186 | 0.2006 | 0.0128 | **0.0020** | 0.0285 | 0.1736 |
>
> The second table shows the fairness metrics (independence (Ind), separation (Sp), and sufficiency (Sf)), along with inaccuracy (Ina).
> | Algorithm | ALM | SGD | SSL-ALM | SSw | StGh | SGD-Fairret |
> | ---- | :----: | :----: | :----: | :----: | :----: | :----: |
> | Ina: mean | 0.2725 | **0.1649** | 0.2651 | 0.4515 | 0.3391 | 0.1724 |
> | Ina: q25 | 0.2641 | **0.1159** | 0.2604 | 0.3953 | 0.2838 | 0.1259 |
> | Ina: q75 | 0.2794 | **0.2084** | 0.2720 | 0.4915 | 0.3840 | 0.2136 |
> | Ind: mean | **0.0551** | 0.1136 | **0.0561** | 0.0865 | 0.0983 | 0.1211 |
> | Ind: q25 | **0.0233** | 0.0663 | **0.0266** | 0.0319 | 0.0386 | 0.0561 |
> | Ind: q75 | **0.0717** | 0.1572 | 0.0807 | 0.1037 | 0.1089 | 0.1624 |
> | Sf: mean | 0.3330 | **0.1428** | 0.3247 | 0.2615 | 0.2933 | 0.1859 |
> | Sf: q25 | 0.1837 | **0.0830** | 0.1724 | 0.1523 | 0.1645 | **0.0860** |
> | Sf: q75 | 0.4703 | **0.1971** | 0.4549 | 0.3681 | 0.4230 | 0.2837 |
> | Sp: mean | 0.1456 | **0.1309** | 0.1341 | 0.1762 | 0.2069 | 0.1574 |
> | Sp: q25 | 0.1055 | **0.0662** | 0.0872 | 0.0975 | 0.1060 | **0.0684** |
> | Sp: q75 | 0.1945 | 0.2110 | **0.1691** | 0.2195 | 0.2727 | 0.2414 |
>
> Overall, the results with the multiple-value attribute confirm the observations made for the binary attribute:
> - Concerning constraint satisfaction, Fairret regularization is better compared to SGD, but is difficult to use to reach a prescribed accuracy. SSw is the method that best meets the constraints, followed closely by SSL-ALM and ALM. Then, Stochastic Ghost ends with twice the constraint value of ALM on average.
> -  Concerning (in)accuracy, SGD performs best (0.16), as expected, and Fairret regularization follows closely (0.17). ALM and SSL-ALM have inaccuracy of (0.27). Then, Stochastic Ghost and SSw have inaccuracies of 0.34 and 0.45. In all of our experiments, these two methods perform poorly in terms of accuracy.
> - Concerning Independence, ALM and SSL-ALM perform best, as for the binary attribute case (see Table 3 in our paper). Fairret has performance comparable to SGD.
> - Concerning Sufficiency, as in the binary attribute case, ALM and SSL-ALM have values twice as large as SGD and Fairret,
> - Concerning Separation, SSL-ALM and ALM have values (0.13 / 0.15) comprised between SGD (0.13) and Fairret (0.16). In this case, these methods have similar performance, whereas for the binary attribute case, SSL-ALM and ALM had values twice as small as SGD and Fairret.
>
> We hope to have addressed all your concerns. We remain at your disposal may you have any further questions or require additional information. We would be grateful if you could consider revising your score based on the answers we provided.

---

> ### Comment · Area_Chair_Vijz · 2025-08-05
>
> Hi Reviewer 2yzk,
>
> Could you please make additional comments on whether and how well you think the authors have answered your questions?
>
> Thanks,
>
> Area Chair.

---

> ### Comment · Reviewer_2yzk · 2025-08-05
>
> I would like to thank the authors for their effort on answering my concerns and I really appreciate the additional experimental results provided.
>
> In my opinion, such discussion regarding the multi-valued attributes and the treatment of several attributes at the same time will further improve the paper and benefit future work.
>
> I will update my score accordingly, since the authors extensively discuss and provide evidence for my main concern.

---

> > ### Author Response · Authors · 2025-08-06
> > **Thanks for reviewing**
> >
> > We are glad that you deemed our revisions convincing and updated your score accordingly. Thank you for the time you invested in reviewing our paper and providing usefull feedback.
> >
> > Best regards,
> > The authors

---

### Official Review · Reviewer_84XC · 2025-07-03

**Rating:** 4
**Confidence:** 2

**Summary:**

This paper presents a comprehensive benchmark and open-source toolbox to evaluate stochastic approximation algorithms for fairness-constrained training of deep neural networks (DNNs), focusing on real-world, large-scale tasks constructed from the US Census dataset. It reviews key theoretical and algorithmic challenges in fairness-constrained training, implements three recently proposed stochastic optimization algorithms, and empirically compares them alongside relevant baselines on a well-defined classification task. The work is positioned to serve both as a framework for practitioners and as a catalyst for future algorithmic research in fairness-constrained deep learning.

**Dataset Code Accessibility:**

Yes

**Ethical Considerations:**

No, there are no or only very minor ethics concerns

**Final Justification:**

I have read the authors' response. They have partially addressed my concerns. They have provided additional experiments to further discuss the adaptability of the benchmark to multiple fairness measurements. However, the discussion of failure modes remains inadequate. I will keep my positive rating.

**Limitations Weaknesses:**

- **Limited Scale**: While the use of ACS/Folktables is meaningful, the empirical evaluation only covers a single binary classification task based on tabular data without other modalities (text, vision), which may restrict generalizability. Besides, the main results focus on a fairly classic binary subgroup split (white vs. non-white). Experiments with more granularity (e.g., multi-group, intersectional, or continuous protected attributes) would strengthen the contribution and demonstrate robustness.

- **Discussion of Failure Modes**: The failures of SSw in Sec. 4.2 need a broader discussion on when or why certain algorithms degrade (e.g., in extremely imbalanced data, high-dimensional features, or small-group splits). Besides, the impact of train-test drift on fairness metrics is less explored, where more in-depth analysis (e.g., statistical significance, calibration plots) could be useful.

**Strengths Contributions:**

- **Motivation**: The authors address a substantial gap in standardizing empirical evaluation for fairness-constrained models. The benchmark built on the US Census (Folktables) ensures strong coverage and potential for wide applicability in fairness research.
- **Implementation**: The authors implement several advanced stochastic approximation algorithms that had not been previously implemented for this task, and open-source the code as a python-package, which significantly enhances reproducibility and accessibility.
- **Experiments**: The authors comprehensively compare fairness-aware baselines to standard ERM, as well as ERM enhanced with fairness regularization, both in terms of optimization and fairness objectives. Their discussion of failure/success modes across algorithms is thorough and concretely substantiated by results tables and figures.

---

> ### Author Rebuttal · Authors · 2025-07-31
>
> We thank you for your careful reading and comments. Here are further details.
>
> > Limited Scale: While the use of ACS/Folktables is meaningful, the empirical evaluation only covers a single binary classification task based on tabular data without other modalities (text, vision), which may restrict generalizability.
>
> The main contributions of this work are the fairness constraint formulation (eq. 4), and the toolbox. They apply to training AI models with arbitrary modalities, either tabular data as we show in the experiments, or text, image, video, sound, etc. We believe that the current experimental section reasonably supports and illustrates these two contributions on tabular data. Providing an exhaustive numerical study on image classification or text generation would only marginally improve the support of these two contributions, while requiring significant computational resources – which are not available to us currently.
>
> > Besides, the main results focus on a fairly classic binary subgroup split (white vs. non-white). Experiments with more granularity (e.g., multi-group, intersectional, or continuous protected attributes) would strengthen the contribution and demonstrate robustness.
>
> We propose below two tables that describe an additional experiment. We consider the same dataset and learning task, with the marriage status (MAR) as protected attribute, that features 5 values. This is modeled by adding one constraint for each pair of values of the protected attribute, resulting in 10 constraints. The algorithms are run with a 3 minutes time budget, and the same hyperparameters as in the paper. Both tables provide the mean / quantile 0.25 / quantile 0.75 of corresponding values for the output model of each algorithm, over 10 runs, on the training set. In both tables, the lowest values in each line are shown in bold. Multiple values are in bold if they are very close to each other.
>
> The first table shows the constraints in the absolute value. We recall that constraints are satisfied when the absolute value is lower than 0.005. In that scenario, only SSw meets the constraints on average. ALM and SSL-ALM are within 180% of the constraint bound.
> | Algorithm | ALM | SGD | SSL-ALM | SSw | StGh | SGD-Fairret |
> | ---- | :----: | :----: | :----: | :----: | :----: | :----: |
> | mean | 0.0131 | 0.1267 | 0.0091 | **0.0016** | 0.0264 | 0.0936 |
> | quantile 0.25 | 	0.0051 | 0.0505 | 0.0032 | **0.0006** | 0.0075 | 0.0344 |
> | quantile 0.75 | 0.0186 | 0.2006 | 0.0128 | **0.0020** | 0.0285 | 0.1736 |
>
> The second table shows the fairness metrics (independence (Ind), separation (Sp), and sufficiency (Sf)), along with inaccuracy (Ina).
> | Algorithm | ALM | SGD | SSL-ALM | SSw | StGh | SGD-Fairret |
> | ---- | :----: | :----: | :----: | :----: | :----: | :----: |
> | Ina: mean | 0.2725 | **0.1649** | 0.2651 | 0.4515 | 0.3391 | 0.1724 |
> | Ina: q25 | 0.2641 | **0.1159** | 0.2604 | 0.3953 | 0.2838 | 0.1259 |
> | Ina: q75 | 0.2794 | **0.2084** | 0.2720 | 0.4915 | 0.3840 | 0.2136 |
> | Ind: mean | **0.0551** | 0.1136 | **0.0561** | 0.0865 | 0.0983 | 0.1211 |
> | Ind: q25 | **0.0233** | 0.0663 | **0.0266** | 0.0319 | 0.0386 | 0.0561 |
> | Ind: q75 | **0.0717** | 0.1572 | 0.0807 | 0.1037 | 0.1089 | 0.1624 |
> | Sf: mean | 0.3330 | **0.1428** | 0.3247 | 0.2615 | 0.2933 | 0.1859 |
> | Sf: q25 | 0.1837 | **0.0830** | 0.1724 | 0.1523 | 0.1645 | **0.0860** |
> | Sf: q75 | 0.4703 | **0.1971** | 0.4549 | 0.3681 | 0.4230 | 0.2837 |
> | Sp: mean | 0.1456 | **0.1309** | 0.1341 | 0.1762 | 0.2069 | 0.1574 |
> | Sp: q25 | 0.1055 | **0.0662** | 0.0872 | 0.0975 | 0.1060 | **0.0684** |
> | Sp: q75 | 0.1945 | 0.2110 | **0.1691** | 0.2195 | 0.2727 | 0.2414 |
>
> Overall, the results with the multiple-value attribute confirm the observations made for the binary attribute:
> - Concerning constraint satisfaction, Fairret regularization is better compared to SGD, but is difficult to use to reach a prescribed accuracy. SSw is the method that best meets the constraints, followed closely by SSL-ALM and ALM. Then, Stochastic Ghost ends with twice the constraint value of ALM on average.
> -  Concerning (in)accuracy, SGD performs best (0.16), as expected, and Fairret regularization follows closely (0.17). ALM and SSL-ALM have inaccuracy of (0.27). Then, Stochastic Ghost and SSw have inaccuracies of 0.34 and 0.45. In all of our experiments, these two methods perform poorly in terms of accuracy.
> - Concerning Independence, ALM and SSL-ALM perform best, as for the binary attribute case (see Table 3 in our paper). Fairret has performance comparable to SGD.
> - Concerning Sufficiency, as in the binary attribute case, ALM and SSL-ALM have values twice as large as SGD and Fairret,
> - Concerning Separation, SSL-ALM and ALM have values (0.13 / 0.15) comprised between SGD (0.13) and Fairret (0.16). In this case, these methods have similar performance, whereas for the binary attribute case, SSL-ALM and ALM had values twice as small as SGD and Fairret.
>
> > Discussion of Failure Modes: The failures of SSw in Sec. 4.2 need a broader discussion on when or why certain algorithms degrade (e.g., in extremely imbalanced data, high-dimensional features, or small-group splits).
>
> We agree that a discussion of when and why algorithms degrade is relevant, however, it is delicate to discuss this at this stage. Indeed, from a theoretical point of view, the algorithms are not sufficiently understood: we are not aware of their convergence guarantees in the general setting of nonsmooth nonconvex optimization, nor of complexity guarantees in that general realistic setting. Besides, from a practical point of view, such a discussion requires extensive computational experiments, including testing against extremely imbalanced data, high-dimensional features, or small-group splits as you suggest. We feel that this is beyond the scope of this work, whose main contributions are to provide fairness constraints formulations, and a toolbox that allows using and testing algorithms easily. It is our hope that this work will generate interest in and facilitate exhaustive comparisons of stochastic constraints algorithms.
>
> Still, we can make the following comment on the specific case of SSw. The performance of SSw heavily depends on the constraint tolerance hyperparameter, and we have searched for a value that strikes a balance between optimizing loss and meeting constraints. That value proved elusive, which is a first indication that SSw does not perform as well as SSL-ALM or Stochastic Ghost, for which the hyperparameters suggested in the references provided good performance and were used directly.
>
> > Besides, the impact of train-test drift on fairness metrics is less explored, where more in-depth analysis (e.g., statistical significance, calibration plots) could be useful.
>
> Figures 3 and 4, and Table 3 provide the train and test values for the four fairness metrics, along with the prediction distribution, thus giving a glimpse of the train-test drift in practice. A study of the statistical drift incurred by estimators obtained by (eq. 4) is of course highly relevant, but pertains more to the field of Statistics, while our focus here is on the optimization performance of the stochastic algorithms (do they converge to solutions of (eq. 4)?). On the topic of train-test drift of accuracy, we believe that relevant or close results may already exist in the field of Distributionally Robust Optimization, or Wasserstein Distributionally Robust Optimization; see e.g. [Universal generalization guarantees for Wasserstein distributionally robust models, Le and Malick, 2025] and references therein. For the other fairness metrics, we are not aware of studies that bound the drift that may be incurred by train-test distribution shift.
>
> We hope to have addressed all your concerns. We remain at your disposal may you have any further questions or require additional information. We would be grateful if you could consider revising your score based on the answers we provided.

---

> > ### Comment · Reviewer_84XC · 2025-08-05
> >
> > Thanks for the authors' response. The response has addressed my concern. I think the current rating is proper.

---

> > > ### Author Response · Authors · 2025-08-08
> > > **Thank you Reviewer 84XC**
> > >
> > > We would like to thank the reviewer for time and effort invested to provide thorough and constructive feedback. We are glad that you found this work interesting.
> > >
> > > Best regards, Authors

---

### Decision · Program_Chairs · 2025-09-18

**Decision:**

Reject

**Comment:**

This paper presents a benchmark and open-source toolbox for evaluating stochastic approximation algorithms in fairness-constrained training of deep neural networks. The reviewers agree that this is an important and timely contribution, filling a critical gap by standardizing empirical evaluation in fairness-constrained optimization. The work is well-motivated, clearly written, and provides reproducible implementations of several advanced algorithms, which will be valuable to both practitioners and researchers.

The strengths highlighted include:

* Addressing a substantial gap in benchmarking fairness-constrained deep learning.
* Implementation of multiple algorithms within a unified, open-source framework.
* Comprehensive comparisons across fairness-aware methods and baselines, offering actionable insights into trade-offs and failure modes.

The main concerns raised relate to the limited scope of experiments (single dataset, binary protected attribute) and relatively narrow evaluations. Reviewers also suggest broader dataset coverage, additional baselines, and more in-depth analyses (e.g., multi-group settings, hyperparameter sensitivity, dataset generalization). If this work is accepted, I suggest the authors try to improve their experiments along some of these directions in the final version.

===== FINAL UPDATE FROM DB Track PCs ====

The final decision for this paper has been taken by the program chairs after consultation with the SACs. All Senior Area Chairs have ranked papers according to the feedback from the AC during the review process. We decided to leave the original meta-review to reflect the opinion of the AC in light of the initial discussions with reviewers and SAC.